# Mid-Latitude Mesospheric Zonal Wave 1 and Wave 2 in Recent Boreal Winters

Yu Shi [1], Oleksandr Evtushevsky [2], Valerii Shulga [1,3], Gennadi Milinevsky [1,2,4,*], Andrew Klekociuk [5,6], Yulia Andrienko [2] and Wei Han [1]

[1] International Center of Future Science, College of Physics, Jilin University, Changchun 130012, China; shiyu18@mails.jlu.edu.cn (Y.S.); shulga@rian.kharkov.ua (V.S.); whan@jlu.edu.cn (W.H.)
[2] Physics Faculty, Taras Shevchenko National University of Kyiv, 01601 Kyiv, Ukraine; evtush@univ.kiev.ua (O.E.); andrienko.yu@univ.kiev.ua (Y.A.)
[3] Department of Millimeter Radio Astronomy, Institute of Radio Astronomy, National Academy of Sciences of Ukraine, 61002 Kharkiv, Ukraine
[4] National Antarctic Scientific Center, Department of Atmosphere Physics and Geospace, 01601 Kyiv, Ukraine
[5] Antarctic Climate Program, Australian Antarctic Division, Kingston 7050, Australia; Andrew.Klekociuk@awe.gov.au
[6] School of Geography, Earth and Atmospheric Sciences, University of Melbourne, Melbourne 3053, Australia
* Correspondence: gmilin@univ.kiev.ua or genmilinevsky@gmail.com; Tel.: +38-050-3525498

**Abstract:** Planetary waves in the mesosphere are studied using observational data and models to establish their origin, as there are indications of their generation independently of waves in the stratosphere. The quantitative relationships between zonal wave 1 and wave 2 were studied with a focus on the mid-latitude mesosphere at 50°N latitude. Aura Microwave Limb Sounder measurements were used to estimate wave amplitudes in geopotential height during sudden stratospheric warmings in recent boreal winters. The moving correlation between the wave amplitudes shows that, in comparison with the anticorrelation in the stratosphere, wave 2 positively correlates with wave 1 and propagates ahead of it in the mesosphere. A positive correlation $r = 0.5$–$0.6$, statistically significant at the 95% confidence level, is observed at 1–5-day time lag and in the 75–91 km altitude range, which is the upper mesosphere–mesopause region. Wavelet analysis shows a clear 8-day period in waves 1 and 2 in the mesosphere at 0.01 hPa (80 km), while in the stratosphere–lower mesosphere, the period is twice as long at 16 days; this is statistically significant only in wave 2. Possible sources of mesospheric planetary waves associated with zonal flow instabilities and breaking or dissipation of gravity waves are discussed.

**Keywords:** zonal planetary wave; polar vortex; mesosphere; stratosphere; major sudden stratospheric warming

## 1. Introduction

The instability of atmospheric motion is investigated using various experimental methods and atmospheric circulation models. Particular attention is paid to the problem of sudden stratospheric warmings (SSWs), which are associated with large disturbances of the zonal flow of the polar regions [1–3]. The observed zonal circulation anomalies in the winter troposphere–stratosphere–mesosphere indicate that they are formed mainly due to variations in the amplitude of planetary waves with zonal wave numbers 1 (wave 1) and 2 (wave 2) [1,4,5]. Propagating upward from the troposphere, wave 1 and wave 2 can decelerate the strong westerly flow in the stratospheric polar vortex region. This can cause changes in vertical wave propagation, triggering an SSW and subsequently the breakdown of the polar vortex [3,4]. The strength and size of the vortex play a critical role in allowing wave activity to penetrate deep into the stratosphere [6] or, in other words, the stratosphere can influence the planetary wave propagation from the troposphere.

Relationships between wave 1 and wave 2 are determined by the 'wave–mean flow' and 'wave–wave' interactions [6–10]. The negative correlation between the amplitudes of zonal wave 1 and wave 2 is well-known, being previously studied in [11–14]. The 'wave–wave' interaction tends to develop in the middle–upper stratosphere and plays a large role in the observed vacillation between wave 1 and wave 2 [9]. This is manifested in the decline of wave 2, which coincides with the amplification of wave 1 [14].

Quasi-stationary planetary waves, which dominate over traveling waves in the northern hemisphere [15] cannot propagate upward through an easterly stratospheric flow [3,16]. However, quasi-stationary zonal wave 1 and wave 2 have been observed in the mesosphere after the reversal to westerly flow during an SSW [4,5,17]. From their absence below 50 km, it can be inferred that an independent source of quasi-stationary waves lies in the mesosphere [17–20].

Nevertheless, mutual relationships between the vertical and latitudinal dynamical coupling in the stratosphere–mesosphere system during SSW events have been studied [17,21]. It has been shown that there is a significant correlation between the amplitude of the planetary wave near 95 km and the polar westerly wind in the stratosphere near 50 km, accompanied by a three-day delay [21]. As for the frequency composition of wave amplitude variations, the presence of periods of 2–5 days [22,23] and 10–12, 15–16, and 20–30 days [19,21] in mesospheric waves has been described.

The mid-latitude zonal planetary waves around 50°N have been recently analyzed [5,24,25]. It was shown that the altitude dependence is not only in the amplitudes of wave 1 and wave 2 [24] but also in the direction of wave migration. Wave 1 migrates westward (eastward) in the stratosphere (mesosphere) in the pre-warming period, and the dominant westward wave 1 exists in the post-warming mesosphere [5]. In the events SSW 2018 and SSW 2019, a similar equatorward shift of the maximum wave amplitude at 10 hPa was found: 60–80°N (wave 1), 50–70°N (wave 2), and 50–60°N (wave 3) [25].

The SSW 2021 was accompanied by an increase in temperature in the stratosphere (a decrease in the mesosphere), accelerating sharply from January 1 and reaching a maximum (minimum) on January 5 [26–28]. Unlike many others, this SSW was a mixed event not easily classifiable as having either a splitting or displacement of the stratospheric polar vortex [27], suggesting the combined effects of zonal wave 1 and wave 2.

This paper is focused on the coupling between zonal planetary wave 1 and wave 2 in the mid-latitude stratosphere and mesosphere during the SSWs in recent winters. Despite significant advances in the study of zonal wave evolution in the winter stratosphere and mesosphere, the relationship between wave 1 and wave 2 remains poorly understood. Therefore, a detailed analysis of the interrelated behavior of these two zonal wave components remains important. The purpose of this work is zonal wave analysis, focusing on the lagged relationships between wave 1 and wave 2, which have not been analyzed in detail in previous studies. The correlation analysis is applied to time series of wave amplitudes in geopotential height to estimate the degree of coupling between waves. In Section 2, data sources and data processing software are described. In Section 3, delay relationship and lagged correlation coefficients between waves 1 and 2 in the stratosphere–mesosphere during the 2020–2021 winter are analyzed. A discussion is given in Section 4, and Section 5 provides a summary and conclusions.

## 2. Materials and Methods

The zonal waves in the mid-latitude atmosphere in winter were analyzed using data from Aura Microwave Limb Sounder (MLS) satellite observations and the Modern-Era Retrospective Analysis for Research and Applications, Version 2 (MERRA-2) reanalysis. In this paper, Version 4.2 of geopotential height (Z) data from Aura MLS [29] is used. The MLS Z data are scientifically useful over the pressure level range of 261–0.001 hPa (about 9–96 km) with a vertical resolution of ~2 km in the stratosphere and ~6 km in the mesosphere. In the case when there were values excluded by data quality criteria, a one-dimensional interpolation method was used.

MERRA-2 is the global reanalysis produced by the Goddard Earth Observing System (GEOS), which covers the period from January 1980 to the present. The horizontal longitude × latitude resolution is 0.625° × 0.5°, the minimum time resolution is one hour, and the vertical stratification is 72 layers from surface to 0.01 hPa [30]. The reanalysis data were obtained from the Goddard Earth Sciences Data and Information Services Center [31].

The amplitudes of zonal wave 1 and wave 2 in geopotential height (Z1 and Z2, respectively) were calculated using Fast Fourier Transform (FFT) analysis [32]. The statistical significance of the correlation coefficient r between Z1 and Z2 was estimated for the 95% confidence level taking into account the lag-1 autocorrelation when determining the effective sample size $N_{eff}$ [33,34].

The variations in Z1 and Z2 are analyzed during SSW events comparing recent winters between 2017–2018 and 2020–2021. The event of the last winter 2020–2021 is considered in more detail since, unlike many others, it is characterized by the splitting and displacement of the polar vortex in the stratosphere, as noted in Introduction [26], and by a pronounced manifestation of wave 1 and wave 2 in the mesosphere, as shown below. A zonal wave analysis was carried out along the 50°N latitude circle, passing through northern Ukraine and northern China, in order to continue the recent studies of mid-latitude SSW manifestations [5,24,25,35]. The polar vortex edge region at 60°N is considered to demonstrate these manifestations at mid-latitude 50°N.

## 3. Results

### 3.1. Mid-Latitude Manifestations of the Major SSW 2021

The MERRA-2 reanalysis shows a sharp deceleration of zonal wind at 10 hPa, 60°N, with maximum weakening from 1 to 5 January 2021 (vertical lines in Figure 1a) and the appearance of weak easterlies in early January. Relative to the MLS climatology of 2004–2020, the stratospheric temperature poleward of 60°N increased by about 15 K (Figure 1b), the zonal wind reversed in early January at 10 hPa, 55–65°N, and the SSW event became defined as major [27,28]. In the ERA5 reanalysis, a major SSW began on 5 January 2021, and the average temperature of the Arctic stratosphere at 10 hPa rose by close to 30 K in under a week before the onset of the SSW [26]. Similar to Figure 1a from MERRA-2 data, the magnitude of the easterlies in ERA5 data was not particularly remarkable for a major SSW, and the strongest easterlies of –9.3 ms$^{-1}$ occurred on 15 January [26].

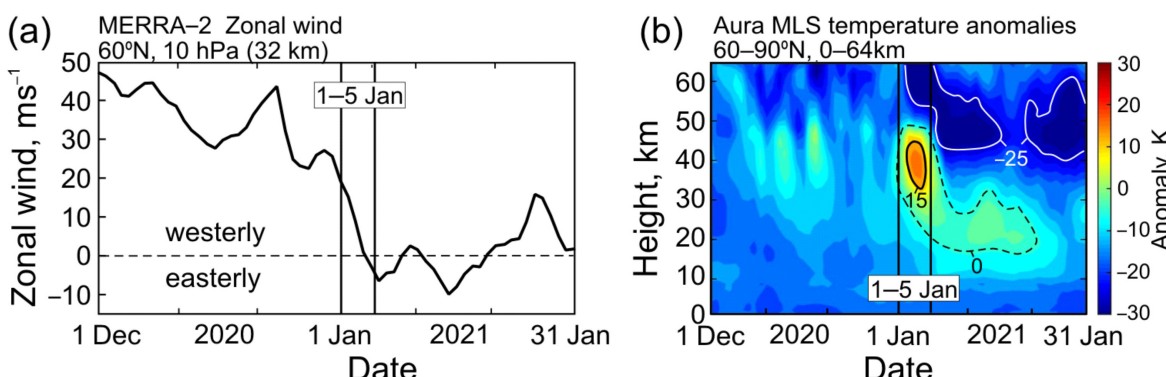

**Figure 1.** (**a**) Zonal wind in the middle stratosphere at 10 hPa (32 km) at 60°N latitude from the MERRA-2 data (dashed horizontal line indicate zero wind) and (**b**) time–height variations of temperature anomalies in the polar cap (60–90°N) from the Aura MLS data relative to climatology 2004–2020. The vertical lines indicate the time interval of 1–5 January 2021 when the SSW event began.

The SSW event was accompanied by simultaneous cooling of about 30 K that occurred in the polar mesosphere (Figure 1b), suggesting distinct height-separated regimes in the stratosphere and mesosphere [27].

Comparing the variability of the amplitudes of wave 1 and wave 2 in the stratosphere (Figure 2) and mesosphere (Figure 3) in recent winters, the same conclusion can be made. The amplitudes Z1 and Z2 varied in time approximately in antiphase in the stratosphere (Figure 2), but they were positively correlated in the mesosphere, with Z1 lagging behind Z2 by several days (Figure 3b,d,f,h). In both cases, wave amplitudes are somewhat larger at the vortex edge (60°N, Figures 2 and 3, left) than at mid-latitude 50°N (Figures 2 and 3, right). This confirms the mid-latitude effect of the off-pole displacement and elongation of the polar vortex due to wave 1 and wave 2 influence, respectively [3,4,24].

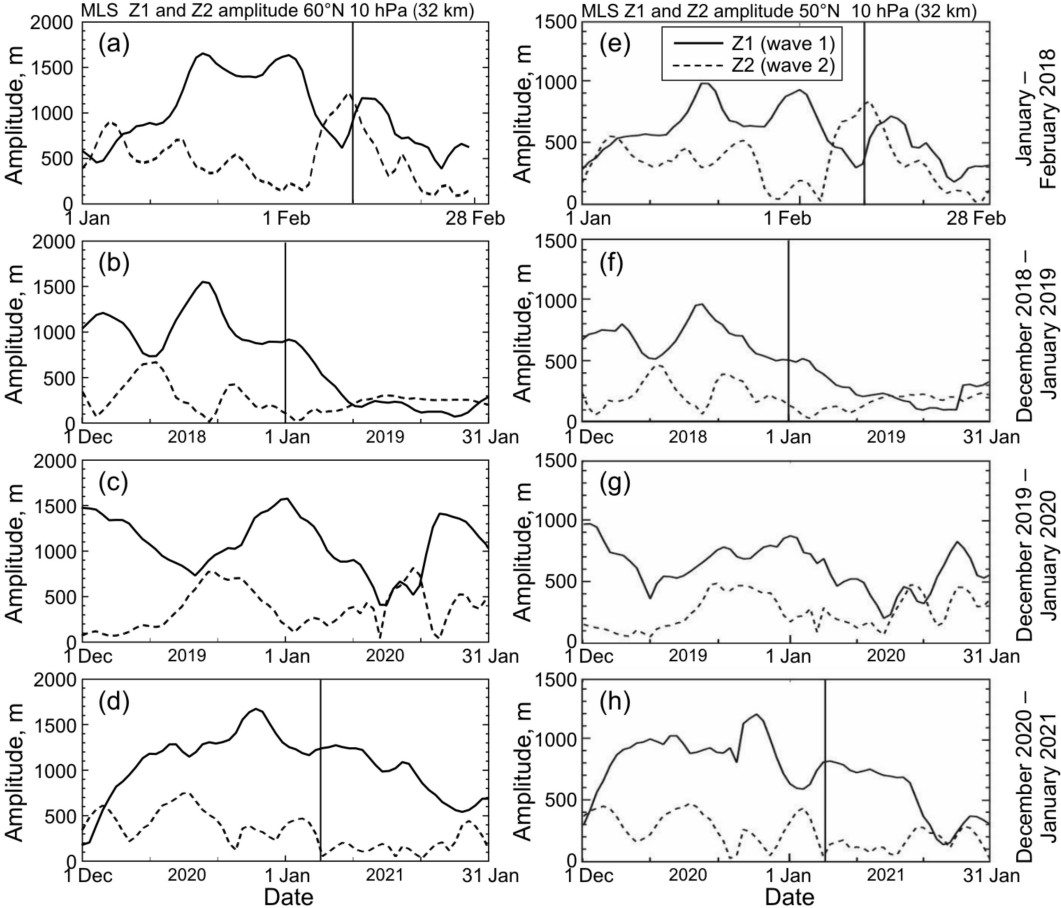

**Figure 2.** Amplitudes of zonal wave 1 (Z1; solid line) and wave 2 (Z2; dashed line) in geopotential height at 10 hPa (32 km), at (**a**–**d**) 60°N and (**e**–**h**) 50°N from the MLS data. Four boreal winters of 2018–2021 are presented. Vertical lines indicate the SSW onset dates.

Unlike zonal wind and temperature (Figure 1), both zonal wave components in the stratosphere showed no noticeable anomalies during the SSW 2021 onset (1–5 January, vertical lines in Figure 2d,h). In the mesosphere at 0.01 hPa (80 km), Z2 exhibited a sharp amplitude peak of about 1000 m on 5 January (dashed curve in Figure 3d,h), just on the date of the strongest easterly in the mesosphere [27]. The wave-1 amplitude, Z1, maximized after the SSW onset (solid curve in Figure 3d,h), which is consistent with the enhancement of mesospheric wave 1 after the zonal wind reversal described in earlier studies [17,19].

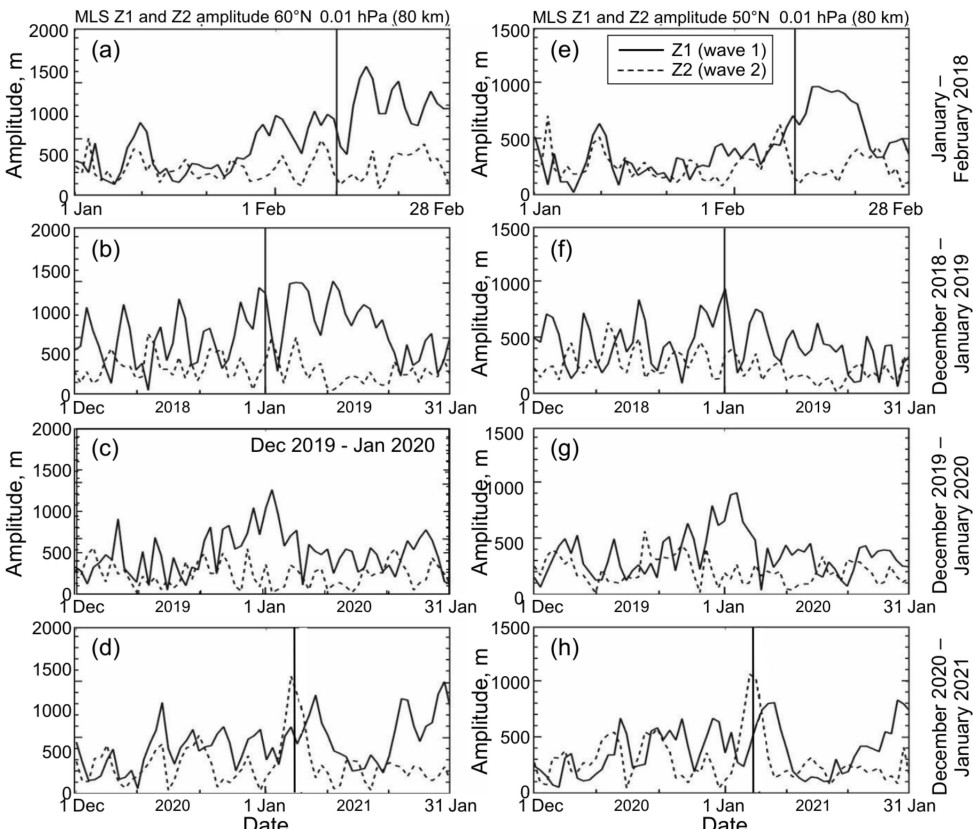

**Figure 3.** Amplitudes of zonal wave 1 (Z1; solid line) and wave 2 (Z2; dashed line) in geopotential height at 0.01 hPa (80 km), at (**a**–**d**) 60°N and (**e**–**h**) 50°N from the MLS data. Four boreal winters of 2018–2021 are presented. Vertical lines indicate the SSW onset dates.

Wave amplitude variations in the stratosphere (10 hPa) and mesosphere (0.01 hPa) in winter 2020–2021 are compared with those in the three preceding winters 2018–2020 in Figures 2 and 3. The anticorrelation between Z1 and Z2, which is clearly visible in the stratosphere at 60°N and 50°N (Figure 2, left and right, respectively), was not observed in the mesosphere, where wave amplitudes varied more frequently in time and they were not so noticeably different in amplitude (Figure 3 and Table A1). The 2-month mean ratio between the wave amplitudes Z1/Z2 in Table A1 decreases upward (e.g., 2.0–3.2 and 1.7–2.5 at 32 km and 80 km, respectively, both at 60°N) and increases poleward (e.g., 1.3–1.8 and 1.7–2.5 at 50°N and 60°N, respectively, both at 80 km). At times, an increase in the amplitude of wave 2 in the mesosphere preceded that of wave 1 (the 2020–2021 winter, Figure 3d,h); this also occurred in the 2018–2019 winter (Figure 3b,f). As far as we know, this effect has not previously been described.

Time–altitude sections of Z1 and Z2 also show that the amplification of wave 2 in the mesosphere (60–80 km) during SSW 2021 leads that of wave 1 by about three days (vertical lines on 5 January and 8 January in Figure 4a,b, respectively). Additionally, as in the case of wave amplitudes in the mid-stratosphere at 60°N (Figure 2, left), there are no noticeable amplitude anomalies at 50°N in early January (the 10-hPa pressure level is shown by black dashed lines in Figure 4a,b). The Z1 and Z2 amplitude variations are even smaller in the lower stratosphere at about 20 km (<300 m, Figure 4, see color scales), which is evidence of relatively low planetary wave activity. This is consistent with the absence of a zonal wind reversal at these altitudes at 55–65°N during the SSW onset [27]. The weak SSW manifestations in the wave 1–2 amplitudes in the lower–middle stratosphere may explain why the SSW 2021 cannot be clearly classified as having a splitting or displacement of the stratospheric polar vortex [27]. In contrast, wave amplitudes at 50°N are much larger in the upper stratosphere/mesosphere and reach 1600 m and 1000 m for Z1 and Z2, respectively

(Figure 4). Easterlies also increase with altitude up to –40 ms$^{-1}$ [27]. Therefore, the SSW 2021 event is mostly developed above 10 hPa pressure level indicated by a black dashed line in Figure 4.

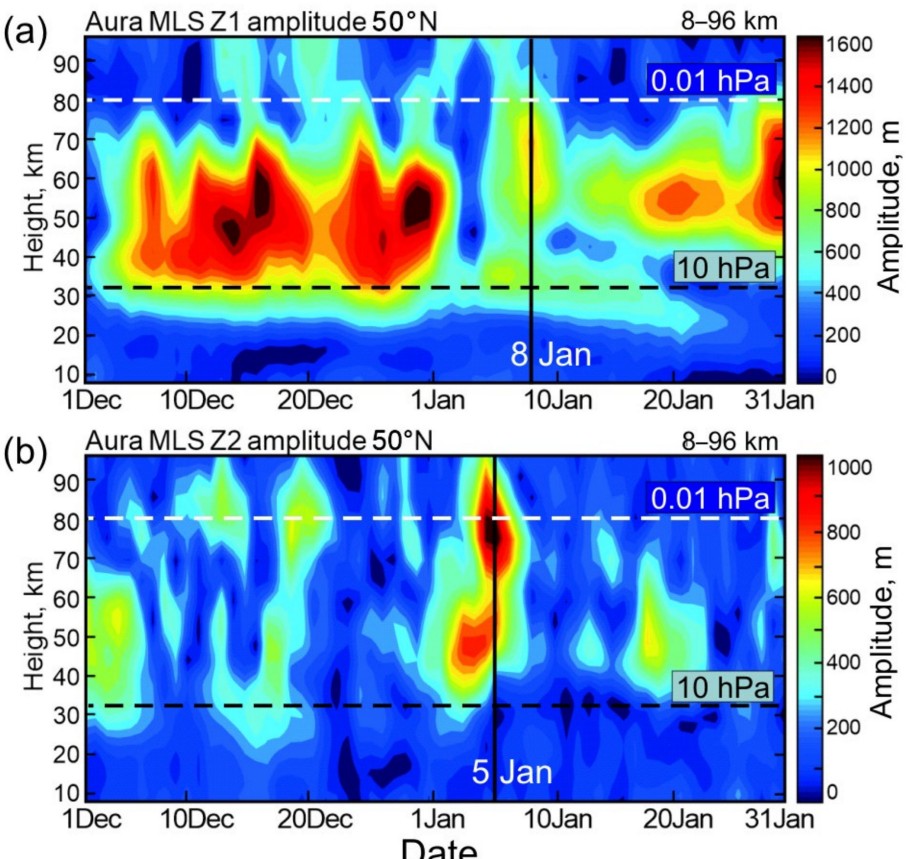

**Figure 4.** The amplitudes (**a**) Z1 and (**b**) Z2 at 50°N in time–altitude section for December 2020–January 2021. Vertical lines indicate the mesospheric wave amplification with wave 1 lagging wave 2 by 3 days.

*3.2. Moving Correlations between Wave 1 and Wave 2*

The difference in the relationship between the amplitudes of wave 1 and wave 2 (Z1 and Z2) in the stratosphere and mesosphere (Figures 2–4) are further analyzed using the moving correlation between them. The linear correlation coefficients are calculated using time lags from –2 days to +13 days. In this paper, we study the lagged relationships between wave 1 and wave 2 on a 1–2-week time scale. A negative lag of –2 days was chosen to see the tendency near zero lag. At the larger negative lags, no significant information was found (not shown). The altitude range covers the atmospheric layer between the lower stratosphere and the mesopause region, 20–96 km (60–0.001 hPa, 34 pressure levels in total according to available MLS Z data).

Figure 5 shows the results of calculating the moving correlation with lagging Z1 relative to Z2 in December 2020–January 2021. A qualitative examination of these results based on the changes in the shapes of the obtained curves shows that there are at least three altitude intervals where moving correlations are of different types. The most distinct is interval 69–96 km (a family of red curves in Figure 5). In this case, no Z1–Z2 anticorrelations typical for the stratosphere (Figure 2) are observed and, on the contrary, a positive correlation (red vertical line in Figure 5) with a maximum of $r = 0.4$–0.6 is found when Z1 lags behind Z2 by 1–5 days. In other words, the variations in Z2 are several days ahead of the variations in Z1, and, due to the relatively high correlative coupling, a change in wave 2 possibly leads to a change in wave 1.

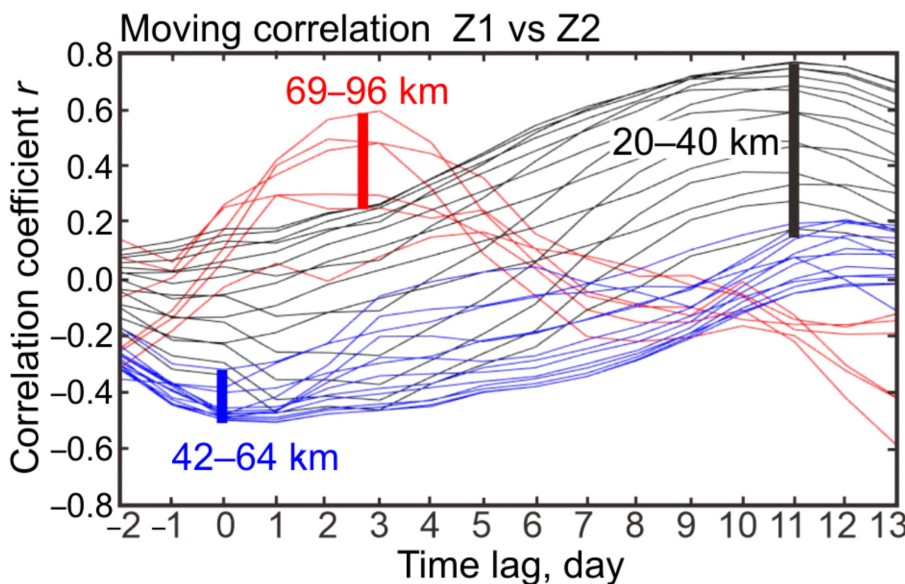

**Figure 5.** Family of moving correlation curves with time lags of the wave 1 amplitude (Z1) relative to the wave 2 amplitude from –2 days to +13 days. The MLS pressure levels between 60 hPa and 0.001 hPa (20–96 km, totally 34 levels) are presented. Three altitude intervals are shown with colored curves: 20–40 km (lower–middle stratosphere, black curves), 42–64 km (upper stratosphere–stratopause–lower mesosphere region, blue curves), and 69–96 km (upper mesosphere–mesopause region, red curves). Vertical lines indicate the strongest correlations at each altitude interval.

The leading role of wave 2 is also seen in the intermediate layer 42–64 km (upper stratosphere–lower mesosphere, blue curves in Figure 5). However, a negative correlation predominates around zero lag (blue vertical line in Figure 5), and the minimum correlation $r \sim -0.4$ appears at l–5-day lags. With a further increase in the time lag, the *r*-values become insignificant (blue curves). A weak negative (strong positive) correlation near zero time lag (lag of 6–13-days) exists in the lower–middle stratosphere in the altitude interval of 20–40 km (Figure 5, black curves). The maximum correlation r = 0.6–0.8 is reached when wave 1 lags relative to wave 2 by 11 days (black vertical line). Therefore, a common feature of the coupling between waves at different altitudes is the time lag in the activity of wave 1 in comparison with wave 2; however, the sign, magnitude, and time lag of the strongest correlation are very different. A particularly noticeable feature is the abrupt transition to a positive lagged correlation 'Z2 vs. Z1' in the upper mesosphere–mesopause region (red curves in Figure 5).

To test the statistical significance of the moving correlation between waves, we first examined a 62-day time series autocorrelation (Table 1) and found an effective sample size for all pressure levels (Figure 6). The lag-1 autocorrelation of times series for wave 1 and wave 2 in the mesosphere is sufficiently large, $r$ = 0.7–0.8 (Table 1). This reduces effective sample size $N_{eff}$ (e.g., [36]):

$$N_{eff} = N\ (1 - r_1\ r_2)/(1 + r_1\ r_2), \tag{1}$$

where $N$ = 62 days is the original time series length, and $r_1$ and $r_2$ are lag-1 autocorrelation coefficients of the two time series being correlated, wave 1 and wave 2, respectively.

**Table 1.** Lag-1 autocorrelation coefficient between wave 1 and wave 2 in the mesosphere.

| Altitude | 75 km | 80 km | 85 km | 90 km |
|---|---|---|---|---|
| Wave1 | 0.70 | 0.77 | 0.78 | 0.75 |
| Wave2 | 0.70 | 0.72 | 0.67 | 0.56 |

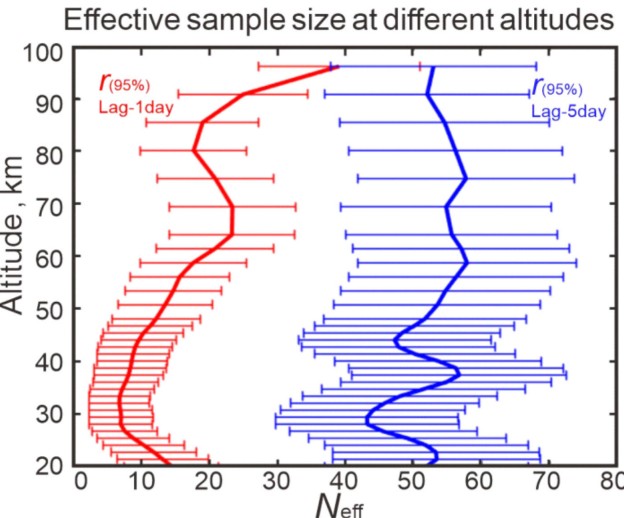

**Figure 6.** Vertical dependence of effective sample size $N_{eff}$ for time series of wave 1 and wave 2 with autocorrelations with time lags 1 day (red curve) and 5 days (blue curve).

In the mesosphere, $N_{eff}$ decreases to 20–30 days with lag-1 autocorrelation but increases to ~55 days with lag-5 autocorrelation (red and blue curves in Figure 6, respectively). As $N_{eff}$ increases with longer lags, the 95% confidence intervals narrow, as shown by the dashed curves in Figure 7.

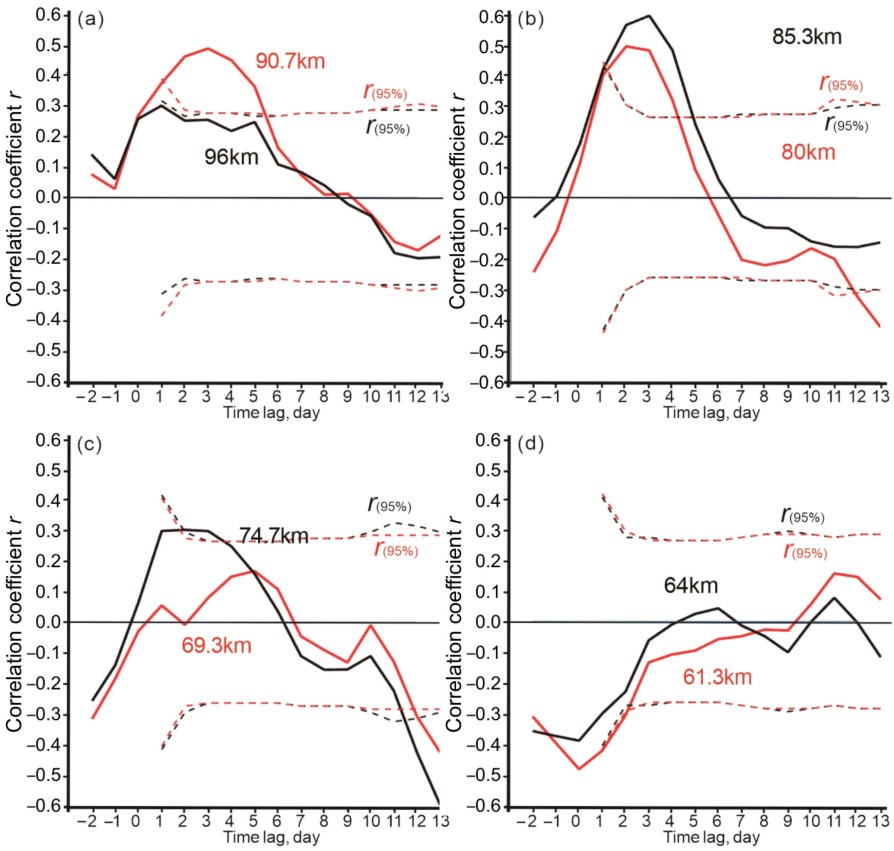

**Figure 7.** Family of moving correlation curves with time lags of the wave 1 amplitude (Z1) relative to the wave 2 amplitude from −2 days to +13 days. The eight MLS pressure levels in the mesosphere–mesopause region between 61 km and 96 km are presented: (**a**) 90.7 and 96.0 km, (**b**) 80.0 and 85.3 km, (**c**) 69.3 and 74.7 km and (**d**) 61.3 and 64.0 km. The 95% confidence limits are shown by dashed curves.

It can be seen that the positive moving correlation at altitudes of 91, 85, 80, and 75 km is statistically significant at the 95% confidence limit when Z2 leads Z1 by 1–5 days and the maximum $r = 0.5$–0.6 is reached at a 3-day leading (Figure 7a–c). The appearance of the positive lagged relationship between wave 1 and wave 2 in the upper mesosphere–mesopause region (Figure 7) quantitatively characterizes their time-dependent interaction and indicates that their origin can be independent of stratospheric waves as noted earlier [17–20].

The statistical significance of the moving correlation in the four consecutive winters 2018–2021 is compared in Figure 8. Each of the three altitude intervals is shown separately, and each curve represents a different pressure level. Totally 16, 12, and 6 pressure levels are presented in altitude ranges of 20–40 km, 42–64 km, and 69–96 km (left, middle, and right columns, respectively, in Figure 8). The 95% confidence limits calculated with lag-1 auto-correlation are indicated by dashed horizontal lines. The course of the moving correlation in individual layers of the atmosphere is very variable and changes from winter to winter. A common feature of the families of curves in Figure 8 is the strong positive correlation ($r = 0.4$–0.8), significant at the 95% confidence limit, which is reached when wave 1 lags behind wave 2 up to least 13 days.

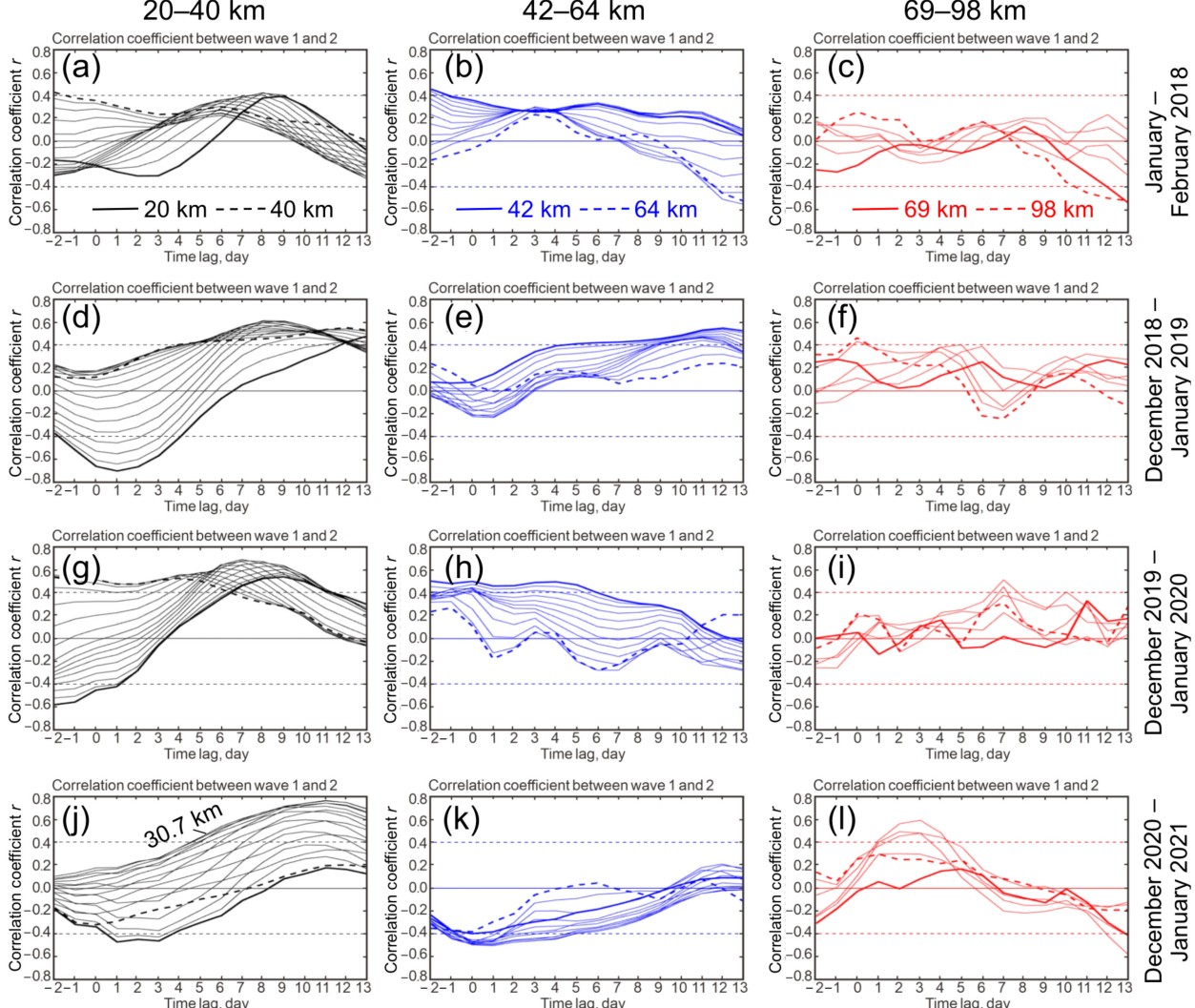

**Figure 8.** Moving correlation between amplitudes of wave 1 (Z1) and wave 2 (Z2) with time lag from −2 days to +13 days. Altitude ranges (left) 20–40 km, (middle) 42–64 km, and (right) 69–98 km are shown. Each curve represents a different pressure level, and the thick solid (dashed) curves show the lowest (highest) levels in each altitude range. From top to bottom: winters (**a–c**) 2018, (**d–f**) 2018–2019, (**g–i**) 2019–2020, and (**j–l**) 2020–2021.

Significant negative correlation is observed mainly in the stratosphere near zero lag (Figure 8, left). It tends to increase with an altitude between 20 km and 40 km to positive values, as seen from the elevation of thick dashed curves relative to thick solid curves in Figure 8a,d,j. The opposite tendency is observed between 42 km and 64 km (Figure 8b,e,h). On the whole, the behavior of the moving correlation in the intermediate layer 42–64 km is unstable: $r$-values increase or decrease as the time lag increases and are statistically significant in some winters at positive or negative levels (Figure 8, middle). No significant altitudinal tendency in the stratosphere–lower mesosphere is seen in the last winter 2020–2021 (compare thick solid and dashed curves in Figure 8j,k), possibly because of sequential elevation–downwelling of the curves relative to about 31 km (Figure 8j). An insignificant altitudinal difference also exists in the upper mesosphere–mesopause region (Figure 8c,f,i,l), where the boundary curves oscillate around zero correlation.

The SSW event does not seem to affect the coupling between waves. For example, winters 2017–2018 (Figure 8a–c) and 2019–2020 (Figure 8g–i) show generally similar tendencies in moving correlations, although a major SSW with zonal wind reversal up to –20 ms$^{-1}$ was observed in the first case, and no SSW and wind reversal occurred in the second case (see 10-hPa zonal winds in Figure A1a). These two winters are characterized by the conditions of the strong stratospheric polar vortices with maximum velocities of 40–60 ms$^{-1}$ (Figure A1a). In conditions of relatively weak vortices in winters 2018–2019 and 2020–2021 (Figure A1b), zonal wind maxima were predominantly at 30–40 ms$^{-1}$ and underwent very similar temporal changes in both cases. Similar are also the main tendencies in moving correlations (Figure 8d–f,j–l, respectively). As in Figure 8l for winter 2020–2021 (see also Figure 5), the maximum positive correlation in the mesosphere in winter 2018–2019 appears at time lag up to 5 days (Figures 8f and A2c,d).

The pairwise similarity of the course of moving correlations under conditions of strong and weak vortices is also noticeable in the stratosphere (Figure 8, left). In the strong vortex events in winters 2017–2018 and 2019–2020, lagged correlation peaks (both positive and negative) appear several days earlier (Figure 8a,g) than in the weak vortex events in winters 2018–2019 and 2020–2021 (Figure 8d,j). The intermediate layer 42–64 km (Figure 8, middle) is characterized by opposite trends in moving correlations in weak and strong vortex events: increasing (Figure 8e,k) and decreasing (Figure 8b,h), respectively.

### 3.3. Periodicity of Z1 and Z2 Variations

As noted in Section 3.1, the wave amplitudes in the mesosphere vary more frequently in time than in the stratosphere (Figures 2 and 3, respectively). The analysis of the periodicity in variations of Z1 and Z2 carried out using the wavelet transform confirms this conclusion. A clear enhancement of shorter 8-day periods in both waves is observed in the upper mesosphere at 80 km (0.01 hPa, top panels in Figures 9 and 10), which are statistically significant in both power spectrum and global wavelet spectrum.

In contrast, the stratosphere–stratopause region (32 km and 48 km, or 10 hPa and 1 hPa) is dominated by longer periods of 15–20 days in wave 1 (Figure 9g,h,j,k, respectively) and ~16 days in wave 2 (Figure 10g–l). If irregular periodicity appears in wave 1 in the stratosphere–lower mesosphere (Figure 9d–l), then ~16-day period persists in wave 2 (Figure 10d–l). Thus, it follows from Figures 9 and 10 that a certain prevailing periodicity is inherent not only in the stratosphere and mesosphere but also in the lower mesosphere and upper mesosphere–mesopause region. The upper mesosphere–mesopause region looks especially isolated in 2020–2021 winter due to strong lagged positive coupling of Z1 relative to Z2 (Figure 8l) and unusual amplification of a fairly stable and strong 8-day period in wave 1 and wave 2 (0.01 hPa in Figures 9a–c and 10a–c). This uniqueness of wave properties suggests the existence of a specific circulation regime in the upper mesospheric layer worthy of attention and further study.

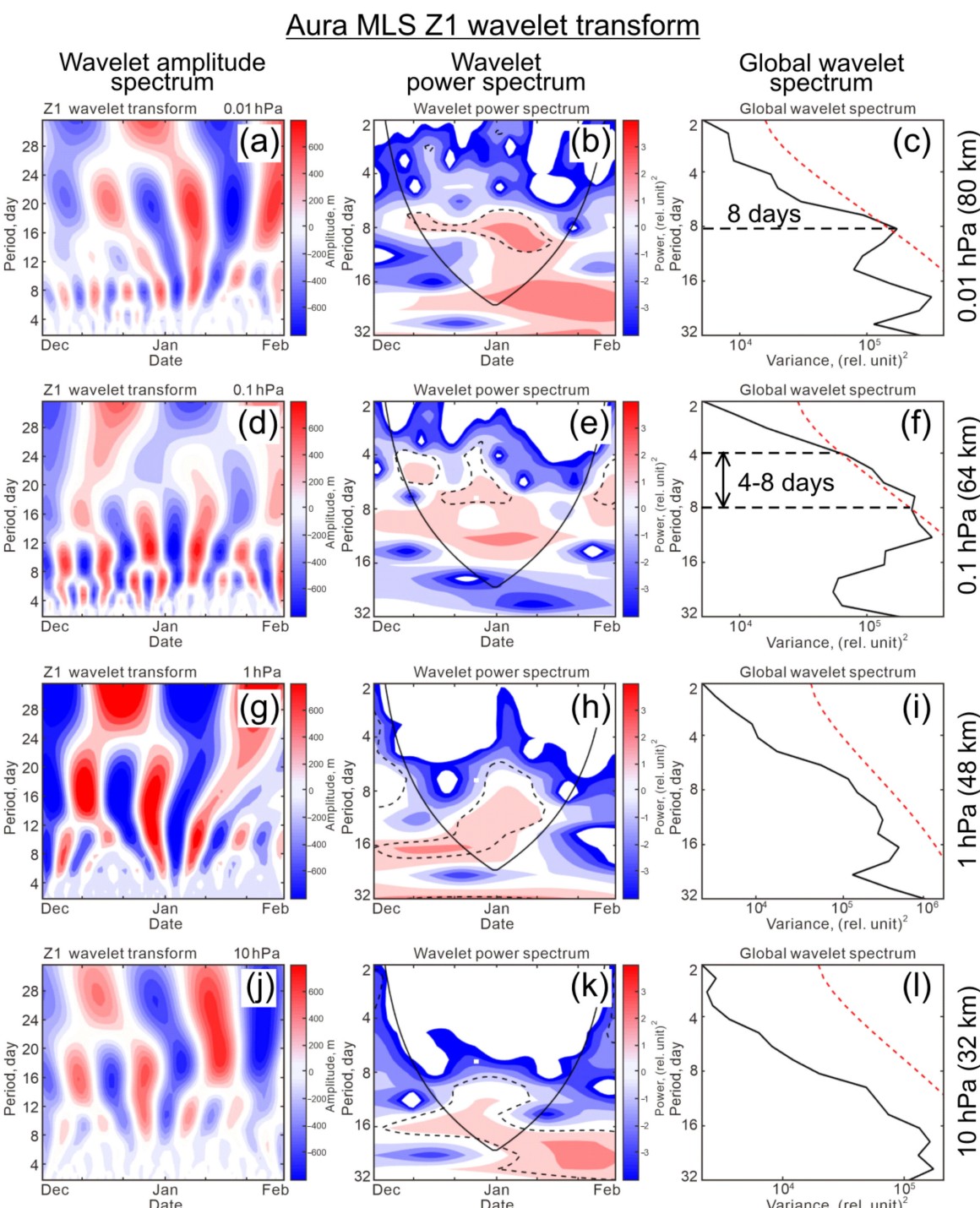

**Figure 9.** Wavelet transforms of time series of wave-1 amplitude, Z1, for (**a–c**) 0.01, (**d–f**) 0.1, (**g–i**) 1, and (**j–l**) 10 hPa. (left) Wavelet amplitude periods, (middle) wavelet power spectrum, and (right) global wavelet spectrum. Dashed contour in power spectrum and dashed line in global wavelet spectrum indicate 95% confidence level. Note that the vertical scale in the left column is the reverse of the scales in the other two columns.

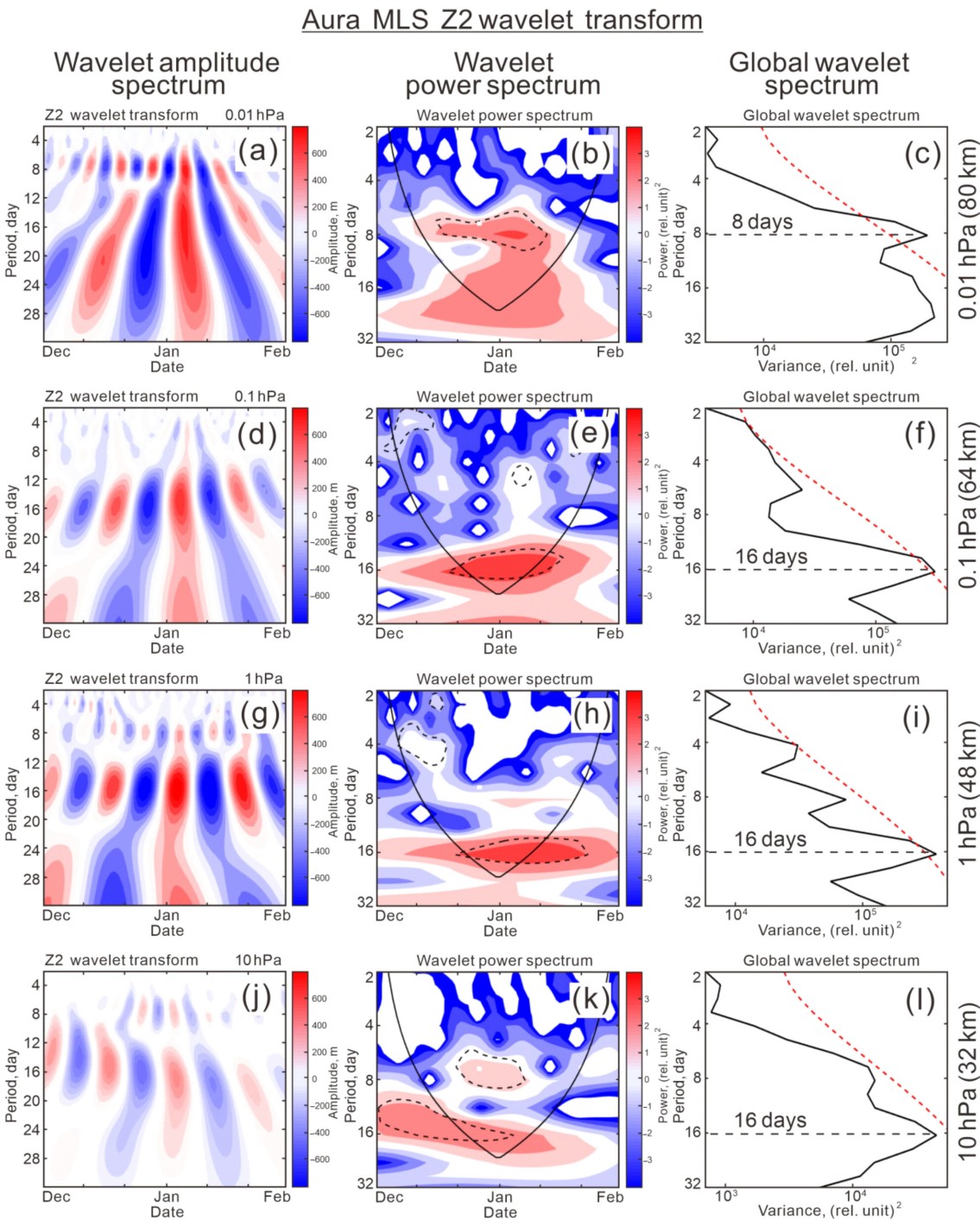

**Figure 10.** Wavelet transforms of time series of wave-2 amplitude, Z2, for (**a–c**) 0.01, (**d–f**) 0.1, (**g–i**) 1, and (**j–l**) 10 hPa. (left) Wavelet amplitude periods, (middle) wavelet power spectrum, and (right) global wavelet spectrum. Dashed contour in power spectrum and dashed line in global wavelet spectrum indicate 95% confidence level.

## 4. Discussion

It is well known that an increase in the activity of planetary waves generated in winter in the troposphere and propagating into the stratosphere can lead to an SSW and rapid polar vortex breakdown [1,2,6,7,12]. The most important are zonal wave 1 and wave 2, with quasi-stationary waves dominating over traveling waves in terms of total energy [15]. Vertically propagating wave 1 and wave 2 can reach the mesosphere accompanying elevated stratopause events when the stratopause is formed at mesospheric

altitudes of about 80 km [4,5,17,21]. However, if strong easterlies during an SSW block upward wave propagation, waves are nevertheless observed in the mesosphere, which indicates their independent mesospheric source [18–20].

The results of moving correlation (Figures 7 and 8) provide evidence that zonal flow strength in the polar region play an important role in coupling between the two main components of zonal wave in the mid-latitude stratosphere and mesosphere. However, the interaction between mesospheric waves 1 and 2 differs significantly from that in the stratosphere and may not depend on SSW events in particular or on the behavior of stratospheric waves in the winter months in general. Differences in wave properties (namely, in behavior over time and in the ratios of amplitudes) are possible indications that waves 1–2 in the stratosphere and mesosphere may be caused by different mechanisms, as discussed in [18–20].

The results from Section 3 reveal some properties of waves 1 and 2 in the mid-latitude winter stratosphere and mesosphere, which were previously little studied. First, moving correlation shows that the variations in wave 1 amplitude follow the variations in wave 2 amplitude with a time lag of up to about two weeks (Figure 8). Especially noticeable lag effects are observed in the winter mesosphere of 2020–2021 (Figures 5 and 7). Second, in contrast to the typically negative correlation in the stratosphere [11–14] (see also Figure 6, left, near zero lag and Figure A1), a positive correlation with maximum $r = 0.4$–0.6 in the winter of 2020–2021 is unexpected (Figures 7 and 8l). Although the correlation is moderate, it is statistically significant at the 95% confidence limit and peaking with a lag of 1–5 days (Figure 7). This time lag indicates the presence of a causal relationship between the waves. Note that there is an insignificant correlation near zero lag. A significant positive correlation is limited to the upper mesosphere–mesopause region (75–91 km) and also appears in the stratosphere, but with a maximum $r = 0.4$–0.8 at the longer lags of 7–11 days (Figure 8, left). Third, a clear 8-day period appears in the variations of Z1 and Z2 in the mesosphere at 0.01 hPa, 80 km (dashed horizontal line in global wavelet spectra in Figures 9c and 10c), whereas a 16-day period exists at the lower altitudes and is significant in wave 2 only (Figure 10f,i,k). Wave 1 does not reveal a regular periodicity in the stratosphere and lower mesosphere (Figure 9d–l).

The enhancement of planetary wave amplitudes in the mesosphere related to SSW events has been previously noted [17,19]. Increased wave amplitudes around the SSW onset dates are seen in our results at 0.01 hPa (80 km) in Figure 3b,d,f,h. However, these amplitude changes are temporary and last less than about 10 days, while statistically significant lagged correlations between Z1 and Z2 have been established using a 2-month time series (Figures 5, 7 and 8). Note that both waves exist before and after the SSW event and their amplitudes often exceed 500 m (Figure 3). Moreover, the similarity of moving correlation behavior in strong vortex winters 2017–2018 (with the SSW event, Figure 8a–c) and 2019–2020 (with no SSW event, Figure 8g–i) is inconsistent with the difference in zonal wind changes in Figure A1a. The latter are distinguished by wind reversal on the solid curve and the absence of wind reversal on the dashed curve. This confirms that SSW events do not have an important role in the generation of waves 1 and 2 in the upper mesosphere nor in the interaction between them.

In addition, 8-day periodicity occurred in December 2021, i.e., in the pre-warming period (Figures 9a,b and 10a,b)). In [19], wavelet power spectra of wave 1 averaged between 55°N and 70°N were obtained for the winter of 2011–2012. They show a dominant periodicity near 16–17 days in the stratosphere (40–50 km) and the appearance of shorter periods < 10 days (3 and 5–7 days) in the upper mesosphere–mesopause region (80–100 km). The authors note that the wavelet spectrum does not show the presence of any short period waves in the stratosphere. This is broadly consistent with the vertical redistribution of the periods in Figures 9 and 10, except that a sharper separation of the 16-day period (stratosphere and lower mesosphere) and 8-day period (upper mesosphere) is observed in wave 2 (Figure 10). Note that [37] suggest that eastward travelling waves may contribute to part of the increased stratospheric wave flux in the pre-warming period of SSWs. However, in the

periods we have examined, quasi-stationary waves provided the dominant contributions in the stratosphere.

Note that unlike the presence of short mesospheric periods in waves 1–2 in Figures 9a–c and 10a–c and in wave 1 in [19], both in geopotential height, they are absent at mesospheric pressure levels in the MLS temperature [5]. This may be due to the fact that geopotential height is sensitive not only to the temperature anomalies at the single pressure level but integrates the temperature–pressure effect in the atmospheric layers below [38] and can give more complete information about the wave spectra.

Abrupt vertical changes in wave relationship (Figure 8) and periodicity composition (Figures 9 and 10) suggest the existence of an independent source of waves in the upper mesosphere–mesopause region compared to the lower atmospheric layers. As shown earlier, mesospheric planetary waves can be generated in situ by breaking or dissipation of gravity waves [18], instabilities in the zonal mean zonal wind [19], or a mixture of gravity wave drag and instabilities [20].

In the absence of SSW effects in the Z1–Z2 relation in the mesosphere, the mean strength of the polar vortex can be an important factor. This follows from the pairwise similarity of the moving correlation behavior (Figure 8) in the strong and weak vortex winters (Figure A1) described in Section 3.2. Therefore, zonal flow strength and some type of zonal flow instability [19] can play a role in the origin of the mesospheric waves 1 and 2 and their observed properties (Figures 4, 5 and 7–10).

On the other hand, the altitudes of the maximum lagged positive coupling between Z1 and Z2 are 75–91 km (Figures 7 and A2), which coincides with the altitudes of the increasing amplitude of wave 1 generated by gravity waves since this process dominates near the mesopause [6,18]. This is an indirect indication of the possible involvement of gravity waves in the generation and interaction of mesospheric waves 1 and 2 in at least three out of four winters (2018–2019, 2019–2020, and 2020–2021, which are the winters of statistically significant moving correlation in Figures 7, 8d–l and A2). Therefore, our results do not exclude the combined action of both mechanisms [20].

The results unambiguously indicate the leading role of variations in the amplitude of wave 2 compared to variations in wave 1 and a two-fold reduction in the period of wave 2 in the mesosphere (8 days) compared to that in the stratosphere (16 days) in winter 2020–2021. The vertical transition from stratospheric to mesospheric wave behavior differs in the relationship between wave 1 and wave 2 (upper stratosphere–lower mesosphere, Figure 8) and in the periodicity of waves (middle mesosphere, Figures 9 and 10). All these features can be important for a better understanding of the generation, propagation, variability, and interaction of planetary wave in the mesosphere and deserve further in-depth study.

## 5. Conclusions

Zonal wave analysis focusing on the lagged relationship between wave 1 and wave 2 in the mesosphere compared to the stratosphere has been performed. Mid-latitude mesospheric manifestations of the major SSW 2021 were analyzed in comparison with those in (i) the stratosphere and (ii) three previous winters of 2017–2018, 2018–2019, and 2019–2020. Variations of the MERRA-2 zonal wind and the amplitudes of zonal wave 1 and wave 2 in the Aura MLS geopotential height (Z1 and Z2) at 50°N were considered. The moving correlation between Z1 and Z2 was used to find the possible time lag between the zonal wave components, and the wavelet transform was applied to the time series to establish the dominant periodicity. The main findings in this work can be summarized as follows:

(i) The time–latitude sections and moving correlations show that Z2 variations lead Z1 variations in the upper mesosphere in the 2020–2021 winter. The correlation coefficient $r = 0.4$–$0.6$, although not very high, is statistically significant at the 95% confidence level at a lead time of 1–5 days, which indicates a causal relationship between the waves. The lead time in the stratosphere in the four recent winters reached up to about two weeks.

(ii) In contrast to the typically negative correlation between wave 1 and wave 2 in the stratosphere near a zero time lag, a positive correlation in the upper mesosphere–

mesopause region (75–91 km) prevails when wave 1 lags by 1–5 days relative to wave 2 in the 2020–2021 winter.

(iii) A clear 8-day period appears in variations of Z1 and Z2 in the mesosphere at 0.01 hPa (80 km), whereas in the stratosphere–lower mesosphere, a 16-day period is significant and only in Z2, and Z1 does not reveal a regular periodicity at these altitudes.

Abrupt changes in wave coupling and periodicity in the upper mesosphere–mesopause region compared to the lower atmospheric layers indicate different wave sources, which is consistent with earlier works. Mesospheric planetary waves can be generated in situ by breaking or dissipation of gravity waves, instabilities in the zonal mean zonal wind, or a mixture of gravity wave drag and instabilities. The results presented suggest a combination of two mentioned mechanisms in the formation of zonal waves 1 and 2 in the mesosphere and give some new quantitative characteristics about the variability and interaction of waves.

**Author Contributions:** Conceptualization, O.E. and G.M.; methodology, O.E. and Y.S.; data acquisition, Y.S. and O.E.; software, Y.S. and O.E.; validation, O.E., A.K. and G.M.; investigation, O.E., G.M., Y.A. and A.K.; writing—original draft preparation, O.E. and G.M.; writing—review and editing, O.E., Y.S., A.K., G.M., Y.A. and V.S.; visualization, O.E. and Y.S.; supervision, G.M.; project administration, G.M., V.S. and W.H. Each author contributed to the interpretation and discussion of the results and edited the manuscript. All authors have read and agreed to the published version of the manuscript.

**Funding:** This research received no external funding.

**Institutional Review Board Statement:** Not applicable.

**Informed Consent Statement:** Not applicable because the study did not involve humans.

**Data Availability Statement:** Datasets analyzed during the study can be found at the Goddard Earth Sciences Data and Information Services Center (GES DISC) websites, https://disc.gsfc.nasa.gov/datasets/ML2GPH_004/summary/, (accessed on 15 July 2021), and https://gmao.gsfc.nasa.gov/reanalysis/MERRA-2/ (accessed on 15 July 2021).

**Acknowledgments:** The authors thank three anonymous reviewers for their useful comments and valuable suggestions that have helped to improve the manuscript. This work was supported in part by Taras Shevchenko National University of Kyiv, projects 19BF051-08 and 20BF051-02; the Institute of Radio Astronomy of the National Academy of Sciences of Ukraine; and the College of Physics, International Center of Future Science, Jilin University, China. This work contributed to the National Antarctic Scientific Center of Ukraine research objectives and contributed to Project 4293 of the Australian Antarctic Program. The MERRA-2 global reanalysis and Aura Microwave Limb Sounder (MLS) data were obtained from the Goddard Earth Sciences Data and Information Services Center.

**Conflicts of Interest:** The authors declare no conflict of interest.

## Appendix A

**Table A1.** The two-month mean amplitudes of wave 1 (Z1, m) and wave 2 (Z2, m) and ratio between them (Z1/Z2, in bold) in winters 2018–2021 in the stratosphere (10 hPa, 32 km) and mesosphere (0.01 hPa, 80 km).

|  | **Jan–Feb 2018** | **Dec 2018–Jan 2019** | **Dec 2019–Jan 2020** | **Dec 2020–Jan 2021** |
|---|---|---|---|---|
| | **Mesosphere (0.01 hPa, 80 km, 50°N) MLS** | | | |
| Z1 | 428 | 436 | 361 | 381 |
| Z2 | 260 | 240 | 212 | 293 |
| Z1/Z2 | 1.6 | 1.8 | 1.7 | 1.3 |
| | Mesosphere (0.01 hPa, 80 km, 60°N) MLS | | | |
| Z1 | 512 | 546 | 377 | 446 |
| Z2 | 240 | 218 | 191 | 258 |
| Z1/Z2 | 2.1 | 2.5 | 2.0 | 1.7 |
| | Stratosphere (10 hPa, 32 km, 50°N) MLS | | | |
| Z1 | 586 | 470 | 623 | 726 |
| Z2 | 355 | 193 | 255 | 238 |
| Z1/Z2 | 1.6 | 2.4 | 2.4 | 3.1 |
| | Stratosphere (10 hPa, 32 km, 60°N) MLS | | | |
| Z1 | 985 | 711 | 1076 | 1054 |
| Z2 | 490 | 267 | 361 | 330 |
| Z1/Z2 | 2.0 | 2.7 | 3.0 | 3.2 |
| | Stratosphere (10 hPa, 32 km, 60°N) MERRA | | | |
| Z1 | 1003 | 691 | 1081 | 1055 |
| Z2 | 482 | 261 | 358 | 326 |
| Z1/Z2 | 2.1 | 2.6 | 3.0 | 3.2 |

Summary statistics: 80 km, 50°N, Z1/Z2 = 1.3–1.8. 80 km, 60°N, Z1/Z2 = 1.7–2.5. 32 km, 50°N, Z1/Z2 = 1.6–3.1. 32 km, 60°N, Z1/Z2 = 2.0–3.2.

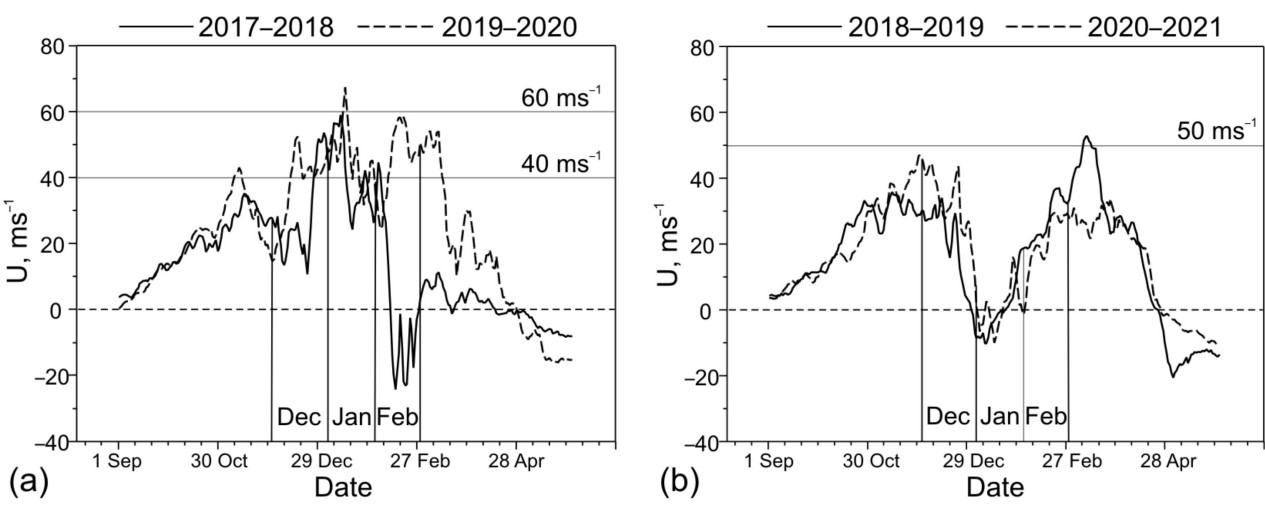

**Figure A1.** MERRA-2 zonal wind at 10 hPa, 60°N, in winters of (**a**) stronger vortex and (**b**) weaker vortex. Note absence of zonal wind reversal in winter 2019–2020 as seen from the dashed curve in (**a**).

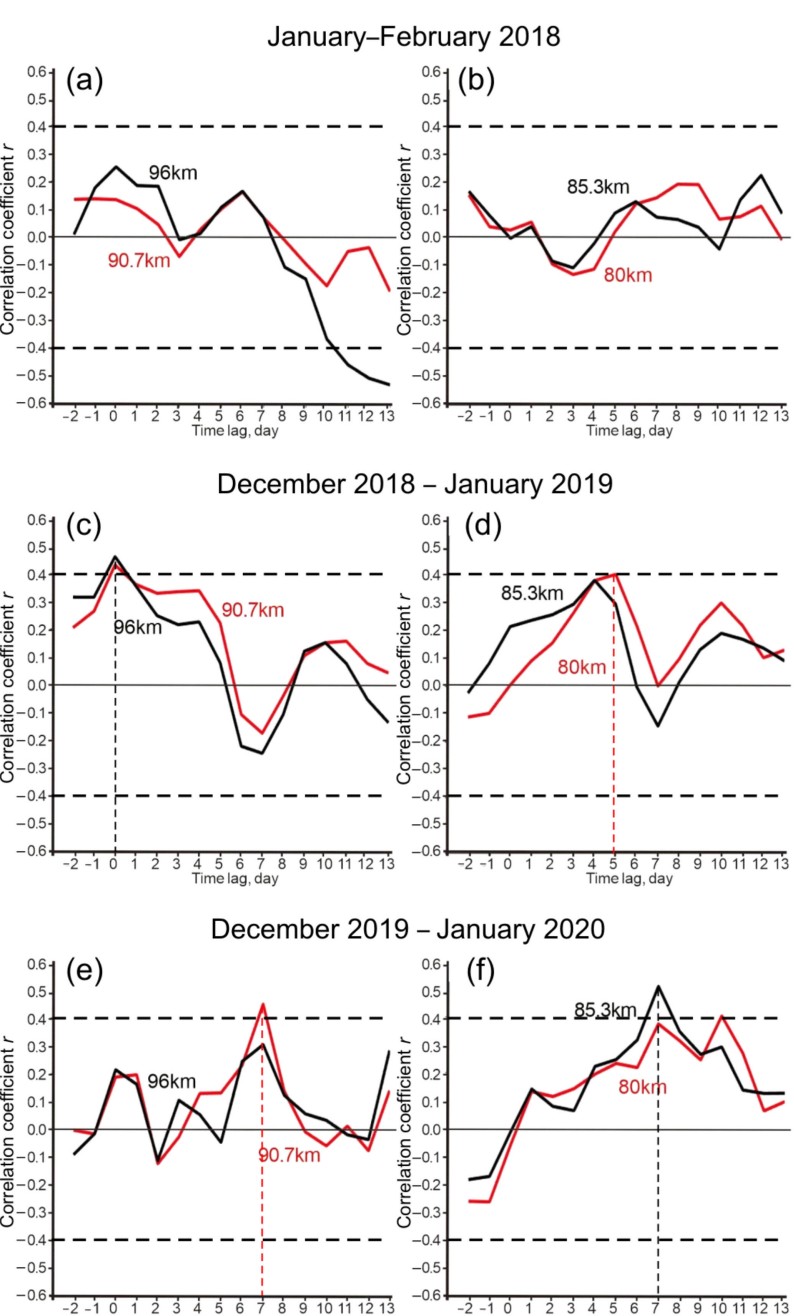

**Figure A2.** Moving correlation between amplitudes of zonal wave 1 (Z1) and wave 2 (Z2) at 50°N in the mesosphere–mesopause region (80, 85, 91, and 96 km). The MLS data for boreal winters of (**a**,**b**) 2018, (**c**,**d**) 2018/2019, and (**e**,**f**) 2019/2020 are presented. Dashed horizontal lines show the 95% confidence limits, and dashed vertical lines indicate maximum correlations reaching or exceeding the positive confidence limit.

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
