# Peer review of "Mid-Latitude Mesospheric Zonal Wave 1 and Wave 2 in Recent Boreal Winters"

_remotesensing, doi:10.3390/rs13183749_

Round 1
Reviewer 1 Report
I found this study is interesting and ready for publications with minor changes.
Below you can find my technical recommendations:
1) The title could be improved, possibly as following "Mid-latitude mesospheric zonal wave 1 and wave 2 in recent boreal winters"
2) line 46
I would suggest to add one sentence on pre-conditioning of SSW:
strength and size of the vortex play a critical role in allowing wave activity to penetrate deep into the stratosphere (see Baldwin et al, SSW, Rev. Geophys. 2020) or in other words the stratosphere can influence the planetary wave propagation from troposphere
3) The following reference could be useful for readers
Baldwin M., Birner T., Brasseur G., Burrows J., Butchart N., Garcia R., Geller M., Gray L., Hamilton K., Harnik N., Hegglin M., Langematz U., Robock A., Sato K., Scaife A. 100 Years of Progress in Understanding the Stratosphere and Mesosphere // Meteorological Monographs. 2019. V. 59. Chapter 27.
4) lines 422-423
Necessary to edit «...three previous winters of 2018, 2019–2019 and 2020», possibly: "... three previous winters of 2017-18, 2018-19, and 2019-20 ..."
Author Response
Response to Reviewer 1 Comments
We thank Reviewer 1 for the useful comments and suggestions that have helped to improve our manuscript.
Point 1. The title could be improved, possibly as following "Mid-latitude mesospheric zonal wave 1 and wave 2 in recent boreal winters"
Response 1. Lines 2–3, we have changed the title to “Mid-latitude mesospheric zonal wave 1 and wave 2 in recent boreal winters”
Point 2. line 46
I would suggest to add one sentence on pre-conditioning of SSW:
strength and size of the vortex play a critical role in allowing wave activity to penetrate deep into the stratosphere (see Baldwin et al, SSW, Rev. Geophys. 2020) or in other words the stratosphere can influence the planetary wave propagation from troposphere
Response 2. Lines 46–49, we added the sentence “The strength and size of the vortex play a critical role in allowing wave activity to penetrate deep into the stratosphere [6] or, in other words, the stratosphere can influence the planetary wave propagation from the troposphere.”
Point 3. The following reference could be useful for readers
Baldwin M., Birner T., Brasseur G., Burrows J., Butchart N., Garcia R., Geller M., Gray L., Hamilton K., Harnik N., Hegglin M., Langematz U., Robock A., Sato K., Scaife A. 100 Years of Progress in Understanding the Stratosphere and Mesosphere // Meteorological Monographs. 2019. V. 59. Chapter 27.
Response 3. We added the reference (Baldwin et al., 2019) as [6] to the text and to the reference list.
Point 4. lines 422-423
Necessary to edit «...three previous winters of 2018, 2019–2019 and 2020», possibly: "... three previous winters of 2017-18, 2018-19, and 2019-20 ..."
Response 4. Lines 116, 451–452, we define the winter seasons using two adjacent years.

Reviewer 2 Report
Review of the manuscript "Mid-latitude mesospheric zonal wave 1 and wave 2 in winter 2020-2021" by Yu Shi et al. submitted to Remote Sensing
The manuscript investigates correlations between mesospheric planetary wave components with zonal wave numbers 1 and 2 and discusses possible mechanisms for their formation. The authors performed zonal wave analysis using geopotential height data, focusing on delays in the relationship between the two components in the mesosphere compared to the stratosphere. I find the research performed interesting and relevant for publication in Remote sensing, however, its presentation should be improved. In my opinion, the goals and the (scientific) conclusions of the paper should be stated more clearly and to the point. The results, discussion and conclusions related sections should be more concise and better organized to avoid repetition. Since there is discussion section, the discussion of the results should be there and not distributed between “Results” and “Discussion”. I recommend a major revision of the manuscript and state specific comments below.
Specific comments
Science related
-
When analyzing the difference in the relationship between the amplitudes of wave 1 and wave 2 in the stratosphere and mesosphere using the moving correlation the temporal delay interval is chosen to be –2 days to +13 days. Why? Please explain the choice and the procedure in more detail.
-
The correlations on the order of 0.4 to 0.6 are not exactly great. The interpretation of the results (in the discussion section) and the conclusions should be made with that in mind.
Wave components
The terminology regarding wave components with wavenumbers 1 and 2 whould be used more concisely and systematically. There is no need to repeat the definition of the wave components under investigation 5 times (in each section except Results) in the manuscript. Furthermore, symbolic notation of the wavenumber (m) is not used anywhere else in the manuscript except the (repeating) definition so I deem it is not needed.
As an example: The amplitudes of zonal wave numbers m = 1 (wave 1) and m = 2 (wave 2) in geopotential height (Z1 and Z2, respectively) were calculated → The amplitudes of zonal waves with wavenumbers 1 and 2 at geopotential height (Z1 and Z2, respectively) were calculated
Please fix these issues.
Heights
The authors present heights in haphazard fashion, sometimes in altitude, sometimes in pressure and sometimes both. Please decide on one of this options. A conversion table in the Appendix could also be a solution.
Figures
The figures in a manuscript should be used as references to confirm or emphasize a certain statement, and not the other way around, where the main manuscript text is explaining them. The captions to all figures should be self-explanatory so that the reader can understand the plut without searching for explanations elsewhere. “As in figure so and so“ is for a caption totally unacceptable.
It is clear that not all graphic material can fit in the manuscript, which should contain only those plots that are needed to emphasize what is written in the text (supplemental figures / tables can be presented in the Appendix). Please reconsider which figures should be where, as frequent referral to figures in the Appendix from the manuscript is very cumbersome for the reader.
-
Why are vertical scales in left column plots if Figs. 7 and 8 inverted with respect to other two columns? Unless there is a good scientific reason for it, it would be better if they were the same.
-
What is the horizontal dashed line in Fig. 1(a)?
Citations and references
The authors are using a mix of two citation styles in the manuscript, namely the style where references are cited according to author names and AIP style, which is generally used for physics, where the numbered references are in brackets. A standard style should be used and I recommend the numbered one. There is no need to reiterate authors’ names in the text (e.g. line 334).
The references should be prepared in a uniformly, in particular, DOI links (the links for [2] and [29] do not work). When referring to web pages, please always add Accessed on (as for [28] and [30]). Please fix this.
Acknowledgements
The URL links to Goddard space center are already in the references, so you may just cite them.
Author Response
Response to Reviewer 2 Comments
We thank Reviewer 2 for the useful comments and valuable suggestions that have helped to improve our manuscript. We corrected the text according to comments and proposals.
General comment
The manuscript investigates correlations between mesospheric planetary wave components with zonal wave numbers 1 and 2 and discusses possible mechanisms for their formation. The authors performed zonal wave analysis using geopotential height data, focusing on delays in the relationship between the two components in the mesosphere compared to the stratosphere. I find the research performed interesting and relevant for publication in Remote Sensing, however, its presentation should be improved. In my opinion, the goals and the (scientific) conclusions of the paper should be stated more clearly and to the point. The results, discussion and conclusions related sections should be more concise and better organized to avoid repetition. Since there is discussion section, the discussion of the results should be there and not distributed between “Results” and “Discussion”. I recommend a major revision of the manuscript and state specific comments below.
Response. We give the main points of work more clearly in “Introduction” (lines 46–49, 87–91) and “Conclusions” (lines 448–449, 461–462). Some text fragments have been moved from “Results” to “Discussion” to avoid repetition (lines 353–361).
Science related
Point 1. When analyzing the difference in the relationship between the amplitudes of wave 1 and wave 2 in the stratosphere and mesosphere using the moving correlation the temporal delay interval is chosen to be –2 days to +13 days. Why? Please explain the choice and the procedure in more detail.
Figure. Moving correlation between amplitudes of wave 1 (Z1) and wave 2 (Z2) with time lag from –2 days to +13 days (top panel, reproduced from Figure 8j–8l in revised manuscript) and, for comparison, with time lag from –13 days to +13 days (bottom panel).
Response 1. In this work, we study the lagged relation between wave 1 and wave 2 on time scale of 1–2 weeks. Negative lag of –2 days was chosen to see the tendency near zero lag. No new information is found at the larger negative lags. This is seen form Figure above (bottom panel). We add short comment in lines 206–209.
Point 2. The correlations on the order of 0.4 to 0.6 are not exactly great. The interpretation of the results (in the discussion section) and the conclusions should be made with that in mind.
Response 2. We take this comment into account in lines 374–377: “Although the correlation is moderate, it is statistically significant at the 95% confidence limit and peaking with a lag of 1–5 days (Figure 7). This time lag indicates presence of a causal relationship between the waves” and lines 460–462: “The correlation coefficient r = 0.4–0.6, although not very high, is statistically significant at the 95% confidence level at a lead time of 1–5 days, which indicates a causal relationship between the waves.”
Wave components
Point 3. The terminology regarding wave components with wavenumbers 1 and 2 would be used more concisely and systematically. There is no need to repeat the definition of the wave components under investigation 5 times (in each section except Results) in the manuscript. Furthermore, symbolic notation of the wavenumber (m) is not used anywhere else in the manuscript except the (repeating) definition so I deem it is not needed.
As an example: The amplitudes of zonal wave numbers m = 1 (wave 1) and m = 2 (wave 2) in geopotential height (Z1 and Z2, respectively) were calculated → The amplitudes of zonal waves with wavenumbers 1 and 2 at geopotential height (Z1 and Z2, respectively) were calculated. Please fix these issues.
Response 3. Corrected to “zonal wave 1 and wave 2” in lines 21, 42–43, 110, 346–347, 452–453.
Heights
Point 4. The authors present heights in haphazard fashion, sometimes in altitude, sometimes in pressure and sometimes both. Please decide on one of these options. A conversion table in the Appendix could also be a solution.
Response 4. We use altitude in km on the vertical axis in Figs. 4 and 6. In the text and in Figs. 9–10, parallel units are used (e.g. lines 100–101: 261–0.001 hPa 96 (about 9–96 km)), because there are readers for whom the ‘pressure surface’ units in hPa are more familiar (used in definition of the SSW and satellite datasets) and readers who are less familiar with such a unit of measurement and it is more convenient for them the height in km. We consider it expedient such a double use of units, without additional reference to the Appendix.
Figures
Point 5. The figures in a manuscript should be used as references to confirm or emphasize a certain statement, and not the other way around, where the main manuscript text is explaining them. The captions to all figures should be self-explanatory so that the reader can understand the plot without searching for explanations elsewhere. “As in figure so and so“ is for a caption totally unacceptable.
Response 5. Full figure captions to Fig. 7 and Fig. 10 are given in revised manuscript.
Point 6. It is clear that not all graphic material can fit in the manuscript, which should contain only those plots that are needed to emphasize what is written in the text (supplemental figures / tables can be presented in the Appendix). Please reconsider which figures should be where, as frequent referral to figures in the Appendix from the manuscript is very cumbersome for the reader.
Response 6. Figures A1 and A2 have been rearranged from Appendix A into the main text as Figs. 2 and 3.
Point 7. Why are vertical scales in left column plots if Figs. 7 and 8 inverted with respect to other two columns? Unless there is a good scientific reason for it, it would be better if they were the same.
Response 7. The left column plots in new Figs. 9 and 10 are shown with the inverted vertical scales.
Point 8. What is the horizontal dashed line in Fig. 1(a)?
Response 8. Caption to Fig. 1a added by the explanation ‘dashed horizontal line indicate zero wind’.
Citations and references
Point 9. The authors are using a mix of two citation styles in the manuscript, namely the style where references are cited according to author names and AIP style, which is generally used for physics, where the numbered references are in brackets. A standard style should be used and I recommend the numbered one. There is no need to reiterate authors’ names in the text (e.g. line 334).
Response 9. We use the ACS style recommended by Remote Sensing Style Guide. Authors’ names in the text have been removed (lines 70, 122 and 346).
Point 10. The references should be prepared in a uniformly, in particular, DOI links (the links for [2] and [29] do not work). When referring to web pages, please always add Accessed on (as for [28] and [30]). Please fix this.
Response 10. DOI links in the revised manuscript are given more accurately: [2] … https://doi.org/10.1175/JCLI-D-17-0648.1 and [30] ... https://doi.org/10.1175/JCLI-D-16-0758.1. Access data have been inserted in [27, 35].
Acknowledgements
Point 11. The URL links to Goddard space center are already in the references, so you may just cite them.
Response 11. The URL links to Goddard Space Center have been removed from Acknowledgements (lines 493–495).

Reviewer 3 Report
In this paper an attempt is made to analyse and discuss the characteristics of teo waves in the mesosphere with the aid of statistical methods. The topic is interesting and the paper is well organised. The results are adequately interpreted. The following comments are suggested:
- abtract, lines 31-32: More information should be presented with respect to possible sources (one or two sentences)
- Introduction, lines 74-80: the objective of the ppaer should be clarified, main;y in relation to previous studies. What is the innovation?
- section 2: more information should be provided for the event of SSW of 2021 (just a brief summary based on other studies. Why this specific event is analysed?)
- section 3.1, lines 121-122: Considering my previous comment, I think that a figure with the anomalies of temperature based on the climatology 2004-2020 should be provided
- Figure 1. Why the latitude 50N and 60 N are specifically selected?
- I think that the number of the figures in the appaendix is large and could be reduced. Furthermore, the remaining figures could be removed in the main manuscript.
Author Response
Response to Reviewer 3 Comments
We thank Reviewer 3 for the useful comments and valuable suggestions that have helped to improve our manuscript. We included in the text of manuscript Reviewer' corrections and proposals.
Point 1. Abstract, lines 31-32: More information should be presented with respect to possible sources (one or two sentences)
Response 1. Possible sources are indicated in lines 31–32: “Possible sources of mesospheric planetary waves associated with zonal flow instabilities and breaking or dissipation of gravity waves are discussed”.
Point 2. Introduction, lines 74-80: the objective of the paper should be clarified, mainly in relation to previous studies. What is the innovation?
Response 2. Lines 76–91, the two last paragraphs in Introduction have been rearranged and the purpose of the paper has been clarified according to the changed title of the article (lines 2–3).
Point 3. Section 2: more information should be provided for the event of SSW of 2021 (just a brief summary based on other studies. Why this specific event is analysed?)
Response 3. The features of the SSW 2021 are briefly noted in lines 116–120.
Point 4. Section 3.1, lines 121-122: Considering my previous comment, I think that a figure with the anomalies of temperature based on the climatology 2004-2020 should be provided
Response 4. The MLS temperature anomalies are shown in new Figure 1b.
Point 5. Figure 1. Why the latitude 50N and 60N are specifically selected?
Response 5. The latitudes 60N and 50N are compared to highlight the polar vortex edge effects at mid-latitude, as indicated in lines 122–123.
Point 6. I think that the number of the figures in the appendix is large and could be reduced. Furthermore, the remaining figures could be removed in the main manuscript.
Response 6. The number of the figures in the Appendix is reduced and the two of them have been moved to Section 3.1 as Figures 2 and 3.

Round 2
Reviewer 2 Report
Review of the revised manuscript "Mid-latitude mesospheric zonal wave 1 and wave 2 in recent boreal winters" by Yu Shi et al. submitted to Remote Sensing
I’m happy with the answers the authors provided and with the revised manuscript, therefore I recommend its publication after a minor revision without an additional review.
A comment regarding the references
I find that the DOI links are still not consistently presented in the references, as in some [4,6,15,17,22,24,25,33,34,35,36] they are presented without a hyperlink as for example in [4]
doi:10.1029/2009GL038586.
and in others [1,2,3,5,7,9,10,11,12,13,14,16,18,19,20,23,26,27,28,30,37] with hyperlink as for example in [2]
https://doi.org/10.1175/JCLI-D-17-0648.1.
Two references [33,36] are none of the above, for example [33]
doi:https://doi.org/10.1175/1520-0493(1983)111<0046:SFSAID>2.0.CO;2.
Please make the references uniform (all DOI should be presented in the same way). I recommend the second option as hyperlinks are very useful to the readers. Also please check that all your hyperlinks actually work (link to the cited papers) - click them to verify them before resubmitting.
Reviewer 3 Report
The authors have responded to my comments and the paper can be accepted for publication